# Social Entrepreneurial Intention and the Impact of COVID-19 Pandemic: A Structural Model

**Inés Ruiz-Rosa [1],[*]** , **Desiderio Gutiérrez-Taño [2]** and **Francisco J. García-Rodríguez [2]**

[1]  Departamento de Economía, Contabilidad y Finanzas, Facultad de Economía, Empresa y Turismo, Universidad de La Laguna, 38071 San Cristóbal de la Laguna, Santa Cruz de Tenerife, Spain

[2]  Departamento de Dirección de Empresas e Historia Económica, Facultad de Economía, Empresa y Turismo, Universidad de La Laguna, 38071 San Cristóbal de la Laguna, Santa Cruz de Tenerife, Spain; dgtano@ull.es (D.G.-T.); fgarciar@ull.es (F.J.G.-R.)

[*]  Correspondence: ciruiz@ull.es

**Abstract:** The interest in promoting social entrepreneurship projects lies in their ability to develop innovative solutions to social and environmental problems. This ability becomes even more important in situations of global crises such as that arising from COVID-19. Based on the Theory of Planned Behavior (TPB), an explanatory structural model of social entrepreneurial intention was tested, and the impact of the COVID-19 crisis on this intention was evaluated. To do this, a quantitative investigation was conducted using a survey of Spanish university students, obtaining a total of 558 responses: 324 before the COVID-19 crisis and 234 during the crisis period (February and June 2020). The results obtained make it possible to validate the explanatory model of social entrepreneurial intention from the perspective of the TPB. In addition, it shows that social entrepreneurial intention decreases in times of deep socioeconomic crises and high uncertainty, such as that caused by COVID-19.

**Keywords:** social entrepreneurial intention; social entrepreneurship; COVID-19; theory of planned behavior

## 1. Introduction

The need to solve social and environmental problems in an innovative way and generate social value is increasingly necessary not only by the public sphere but also by private initiatives [1–3]. In this sense, Horne et al. [4] find that entrepreneurship has great potential to contribute to the Sustainable Development Goals (SDGs).

In this sense, social and environmental entrepreneurship refers to an enterprise project whose objective is to solve a social and/or environmental problem. Moreover, Maer and Noboa [5] understand social entrepreneurship as a process that involves people (social entrepreneurs) who show a tendency toward a specific type of behavior (social entrepreneurship behavior) and who try to carry out that behavior to achieve a tangible result (a social enterprise). The application of the talent, experience, and resources of entrepreneurs in solving social and environmental problems has become a great competitive advantage in many countries [6,7].

Despite growing interest in the concept of social entrepreneurship [8], there is still no clear academic consensus regarding the conceptual delimitation of the term itself, as well as the most appropriate theoretical approach for its analysis, including its antecedents [1,9–12]. Likewise, taking into account the importance of this type of entrepreneurship to mitigate the consequences of economic crises [13], it is necessary to know how such situations affect the behavior or social entrepreneurship intentions of individuals.

Considering the above, this study had a dual objective. On the one hand, to delimit an explanatory structural model of social entrepreneurial intention, analyzing the relationships between this variable and its antecedent variables. On the other hand, the study also aimed to test the effect of a socio-economic crisis with a high level of uncertainty, such as that posed by COVID-19 on social entrepreneurial intention.

To do this, this study first defined the concept and scope of social entrepreneurship. Next, the perspective of the Theory of Planned Behavior (TPB, hereafter) was used to try to delimit the formation of social entrepreneurial intention [14,15]. This theory proposes that entrepreneurial intention depends on the influence that three variables have on it: personal attitude, subjective norms, and perceived behavioral control [15].

The structural model defined was thus empirically tested using a sample of university students, as this population is considered one of the most sensitive to the development of social entrepreneurship projects [9]. The TPB was applied to social and/or environmental entrepreneurial intention, and the model was analyzed for two different periods: before the COVID-19 crisis, and during the pandemic period. A quantitative study was conducted using a survey of university students (University of La Laguna, Spain) obtaining a total of 558 responses: 324 before the COVID-19 crisis and 234 during the crisis period (February and June 2020). This allowed us to analyze whether the crisis had had a positive or negative impact on the social and/or environmental entrepreneurial intention of the sample.

This paper is organized as follows. In the next section, we present the theoretical context and the hypotheses to be tested in relation to the characterization of social entrepreneurial intention and the relation between crises and social entrepreneurial intentions. We subsequently describe the research model as well as sample selection and data collection before reporting the main results. Finally, we conclude with a discussion of results, theoretical and practical implications, limitations, and our main conclusions for further research.

## 2. Theory and Hypotheses

### 2.1. TPB and Social Entrepreneurial Intention

There is a growing interest, both by academic and government institutions, in promoting social and/or environmental entrepreneurship [3,5,11,16,17]. This is justified because these entrepreneurial projects can provide solutions to social and environmental problems in ways that are often more efficient and sustainable than those developed by the public sector [18]. Likewise, Ferri and Urbano [2] state that the social and/or environmental problems emerging in many countries, both developed and developing, have increased the importance of social and/or environmental entrepreneurship as an option to generate social value through social innovation [19,20].

However, despite this growing interest, there is still no clear academic consensus regarding the concept of social entrepreneurship and how to identify and measure it [1,9–12]. In this work, the concept of social entrepreneurship requires social and/or environmental entrepreneurs. The motivation of these entrepreneurs plays a fundamental role. Thus, while traditional entrepreneurship aims to generate profits, social entrepreneurship aims to solve a social and/or environmental problem [5,11,21]. These entrepreneurs are motivated by a strong desire to generate social value [5,22], they are able to identify opportunities focused on solving social and/or environmental problems [23,24] and, therefore, have a collective, and not an individualistic, view of reality [25]. This same author [25] introduces the term "sustainable social value". This concept of sustainability refers to the intention to maintain social activity over time, which, in turn, requires generating business activity with the aim of guaranteeing financial sustainability [26].

In short, social and/or environmental entrepreneurship projects are hybrid models [1,19,27–29] that function like traditional companies but incorporate an objective of a social and/or environmental nature [30]. Since social entrepreneurs are facing social and/or environmental challenges, it is important to understand the variables relating to social entrepreneurial intentions in order to stimulate those

variables [31]. For this reason, it is a priority to know the formation process of social entrepreneurial intention, since it constitutes the previous step to the implementation of any entrepreneurship project and is the best predictor of actual entrepreneurship [14,15,32,33].

Along these lines, the TPB [14,15], used by Krueger and Carsrud [34] to build their entrepreneurial intention model, has become the model that best describes the entrepreneurial process [35] to the degree that it explains the entrepreneurial intention from the interaction, precisely, between personal and social factors. This same model has also been used by Forster and Grichnik [36] to explain the formation of social entrepreneurial intention. Prieto [37] defines social entrepreneurial intention as the purpose that a certain person manifests in starting a social company with the aim of generating social value through innovation.

The TPB proposes that entrepreneurial intention depends on the influence that three variables have on it: personal attitude, subjective norms, and perceived behavioral control [15]. Using this model, multiple academic studies have been carried out that try to analyze the formation of entrepreneurial intention and the relationship with its antecedents (e.g., [38–42]).

In this sense, personal attitude, according to the TPB [14], will depend on the assessment, positive or negative, that a certain person has in relation to the possibility of developing an entrepreneurial project. Indeed, there are several studies that find a positive relation between attitude and entrepreneurial intention (e.g., [39–41]). In our case, as we are analyzing social entrepreneurship, this assessment will be related to social entrepreneurial intention, thus it would be logical to think that:

**Hypothesis 1 (H1).** *There is a positive relationship between social entrepreneurship attitude and social entrepreneurial intention.*

On the other hand, subjective norms refer to the perceived social pressure to carry out, or not, a certain behavior, therefore this element becomes the main reflection of social and cultural values. An estimation of the subjective norm is obtained from the analysis of two variables: the beliefs about how other significant persons think that the individual should behave (normative beliefs) and the motivation that refers to the general tendency that exists in complying with the norms of a group taken as a reference [43]. In this sense, there is a diversity of results when it comes to justifying the relationship between subjective norms and entrepreneurial intention. While some studies have found a significant relation between both [44,45], others have not obtained any relation [46,47].

However, it would be reasonable to expect a positive relationship between this variable and social entrepreneurial intention to the extent that we understand that entrepreneurs are affected by the opinions of people linked to their closest environment about their social entrepreneurial intentions [48]. Following this reflection, the second hypothesis is proposed.

**Hypothesis 2 (H2).** *There is a positive relationship between subjective norms and social entrepreneurial intention.*

Finally, perceived behavioral control refers to the greater or lesser difficulty that a person perceives in performing the action in relation to his or her abilities to control the behavior [49]. Perceived behavioral control is linked to the self-perception of personal abilities and, therefore, is associated with the concept of self-efficacy [50]. In this sense, Smith and Woodworth [51] recognize that a person with a high social entrepreneurial self-efficacy will tend to act more persistently in their goal of creating social value. Therefore, it can be understood that the self-perception of the personal capacity to perform a certain action significantly influences the intention to do that action [44,47]. Following this logic of reasoning, the third hypothesis of this work is proposed.

**Hypothesis 3 (H3).** *Perceived behavioral control positively influences social entrepreneurial intention.*

Finally, following Heuer and Liñán [52] and Liñan and Santos [53], it can be understood that subjective norms represent a form of social capital that could influence the perception of the

entrepreneurial person's personal attitudes and perceived behavioral control. It is for this reason that it seems logical to assume that there could be a positive relationship between subjective norms and social entrepreneurship attitude and between subjective norms and perceived behavioral control linked to the perception of the personal capacity to develop a project [35,48]. To measure these relationships, the fourth and fifth hypotheses of this paper are proposed.

**Hypothesis 4 (H4).** *There is a positive relationship between subjective norms and social entrepreneurship attitude.*

**Hypothesis 5 (H5).** *There is a positive relationship between subjective norms and perceived behavioral control.*

*2.2. Crises and Social Entrepreneurship*

For Hundt et al. [54], the intention to start up an entrepreneurial project is conditioned, in addition to individual characteristics, by the conditions of the economic context, an aspect that must be taken into account to explain entrepreneurial intention, as well as its antecedents [55].

On the other hand, the promotion of entrepreneurship is one of the measures usually considered as a response to situations of economic crises [9], although according to the results obtained by Devece et al. [55], entrepreneurship out of necessity in situations of economic recession is less effective than that arising from the recognition of opportunities. In this sense, Aparicio et al. [56] find a positive relationship between the generation of entrepreneurial projects by opportunity and the economic growth of a given territory. This is why the development of entrepreneurial projects that take advantage of opportunities generates regional economic growth that is greater than that of entrepreneurship for necessity, since, while the latter is limited to solving short-term problems, opportunities can have a long-term impact [55].

Therefore, it would be essential in periods of recession to promote the creation of new businesses, focused on identifying opportunities, with the aim of encouraging economic activity [57]. In the special circumstances of the impact of the COVID-19 crisis, Maritz et al. [58] recognize that entrepreneurs will be key figures during and after the health crisis. Along these lines, [59] point out that the health requirements arising from the pandemic have facilitated the emergence of new business opportunities such as flexible manufacturing, online education, food safety, emergency management, analysis of medical care, care of the elderly, interest in healthy living, telemedicine, cultural services, adaptation of supply chains, remote communication, entertainment or fitness platforms, and the design of smarter cities, among others. Many of these business options could become opportunities for the development of social entrepreneurship projects, in accordance with the definition given above, which focus on solving social and/or environmental problems [23,24].

In short, fostering social entrepreneurship in these circumstances becomes a fundamental tool for generating social and/or environmental change [13]. However, according to the results obtained by Hundt et al. [54] when analyzing the impact of the 2008/2009 crisis on entrepreneurship, the context in which the entrepreneurs find themselves can affect their behavior. Indeed, among the conclusions obtained from the Global Entrepreneurship Monitor project [60], the rate of nascent entrepreneurship decreased notably during the period of 2008-2010 in the countries most affected by the crisis. Devece et al. [55] also conducted a comparative study of new company creation in Spain between 2005 and 2007 and 2008 and 2010, observing that the number of new companies created fell from 400,000 in the first period to 300,000 in the second. Therefore, following Arrighetti et al. [61], the perception of a crisis as a barrier negatively affects entrepreneurial intention, which leads to the last hypothesis of this work:

**Hypothesis 6 (H6).** *Social entrepreneurial intention is lower during COVID-19 than before.*

## 3. Empirical Study

### 3.1. Research Model

Based on what was stated in the previous section, we tested, on the one hand, the suitability of a structural model of social entrepreneurship based on Ajzen's TPB [14] and, on the other hand, the impact of the COVID-19 crisis on social entrepreneurial intention.

In this sense, the construct "entrepreneurial intention" was conceptualized as a latent variable depending on three others: the attitude toward social entrepreneurship, subjective norms, and perceived behavioral control. Finally, the crisis variable COVID-19 was added to the model.

Thus, our research model includes five factors (see Figure 1): social entrepreneurship attitude, subjective norms, perceived behavioral control, social entrepreneurial intention, and crisis variable COVID-19. Each factor was measured with multiple items. All items were adapted from extant literature to improve content validity.

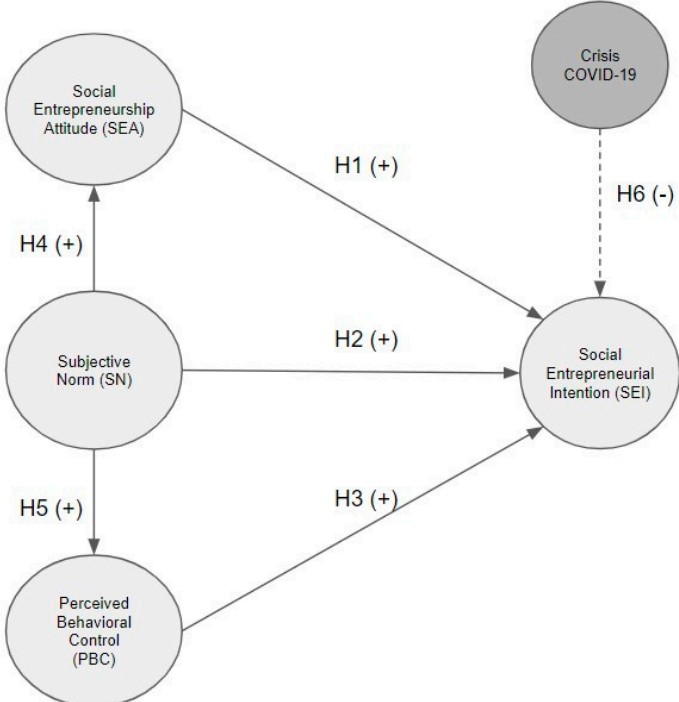

**Figure 1.** Research model.

### 3.2. Measures

A quantitative research design was used for this study through a survey of university students, as, according to the Global Entrepreneurship Monitor (GEM) Report on Social Entrepreneurship [62], this type of entrepreneurship is closely associated with young change-makers, who are idealistic in nature. In fact, the results of the GEM report show there is a greater representation of incipient social entrepreneurs than commercial entrepreneurs among young people between 18 and 34 years old. More specifically, according to Capella-Peris et al. [9], higher education students are one of the most relevant populations for the development of social and/or environmental projects.

The questionnaire developed for the study was structured in three parts. The first part introduced the context. The second part included the items of the constructs of the proposed model, which were measured by a 7-point Likert scale regarding the level of agreement (1 = Strongly disagree to 7 = Strongly agree). In the last part of the questionnaire, classification data were collected: gender, age, studies, and academic year. We included a definition of social and/or environmental entrepreneurship projects

in the survey. We proposed that there could be hybrid models that function like traditional companies but incorporate an objective of a social and/or environmental nature.

The questionnaire was sent to the students by their teachers in February 2020 and in May 2020.

Regarding the items used (Table 1), Armitage and Conner [63] propose three different approaches to measure entrepreneurial intention. One of them is based on the desire to perform an action, another on the probability of performing the action, and the last one centered on the intention to perform said action. These authors corroborate that the latter is slightly more efficient in predicting behavior. To measure social entrepreneurship attitude and subjective norms, the scales proposed by Liñán and Chen [49] were used. The items used to measure social entrepreneurship attitude are supported both by affective considerations (developing social and/or environmental entrepreneurship projects can be an attractive activity that could generate satisfaction) and other more objective aspects. Those linked to subjective norms refer to the perception that entrepreneurs may have about what people close to their environment (family, friends, colleagues) think about their interest in developing a social and/or environmental entrepreneurship project. Finally, to measure perceived behavioral control, the six items proposed by Zhao et al. [64] were used; they refer to the entrepreneur's ability to identify opportunities, offer new products and services, manage projects, and have contact networks and leadership skills.

**Table 1.** Construct and associated items.

| SEI | Social Entrepreneurship Intention |
|---|---|
| | Indicate your level of agreement with the following sentences |
| SEI1 | I am willing to do anything to start a social project. |
| SEI2 | My professional goal is to become a promoter of social projects. |
| SEI3 | I am determined to create a social project in the future. |
| **SEA** | **Social Entrepreneurship Attitude** |
| | Indicate your level of agreement with the following sentences |
| SEA1 | Being a social entrepreneur has more advantages than disadvantages for me. |
| SEA2 | A career as a promoter of social projects is attractive to me. |
| SEA3 | Promoting social projects would be a great satisfaction for me. |
| **SN** | **Subjective Norms** |
| If you decided to create a social project, would people in your close environment approve of that decision? | |
| SN1 | Your closest family. |
| SN2 | Your friends. |
| SN3 | Your study partners. |
| **PBC** | **Perceived Behavioral Control** |
| To what extent do you agree with following statements regarding your entrepreneurial abilities? | |
| PBC1 | Identify new opportunities. |
| PBC2 | Create new products and services. |
| PBC3 | Apply my personal creativity. |
| PBC4 | Be a leader and communicator. |
| PBC5 | Create a network of professional contacts. |
| PBC6 | Successfully organise/manage a project. |

Scale 1 to 7 (1 = Strongly disagree to 7 = Strongly agree).

### 3.3. Sample Selection and Data Collection

Data were obtained from the same students in two phases: the first one prior to the COVID-19 crisis in February 2020 and the second one in full crisis, in June 2020. The information was collected through a self-completed online questionnaire using the Lime Survey platform (version 3.6.3). Students at the University of La Laguna (Spain) were sent the links to the online questionnaire twice in each phase by email. These students belong to the following degrees: Building and Civil Engineering, Social Work, Industrial Relations, and Accounting and Finance. A total of 558 responses were obtained:

324 (58%) before the crisis (response rate 21%) and 234 (42%) responses in the crisis period (response rate 15%).

To verify that the sample size was sufficient, G*Power [65] was used, which suggests that for the test of the proposed model (Figure 1), a minimum sample of 129 individuals is required for a statistical power of 0.95. Therefore, it can be safely concluded that the sample size used was much larger than required for the purposes of this study.

Table 2 shows the profile of the respondents. Most of the responses obtained corresponded to women, 66.3%, and the largest number of questionnaires were completed by first-year students and the least in the last year of the degree, both of which correspond to the distribution of the analyzed population.

**Table 2.** Profile of respondents.

| Gender | Total | Before (February) COVID-19 | During (June) COVID-19 |
|---|---|---|---|
| Female | 66.3% | 67.6% | 64.5% |
| Male | 33.7% | 32.4% | 35.5% |
| **Degree studies** | | | |
| Building and Civil Engineering | 26.6% | 23.7% | 30.6% |
| Social Work | 33.3% | 36.1% | 29.3% |
| Industrial Relations | 18.4% | 19.9% | 16.4% |
| Accounting and Finance | 21.7% | 20.3% | 23.7% |
| **Academic year** | | | |
| 1st | 54.3% | 54.9% | 53.4% |
| 2nd | 19.5% | 20.7% | 17.9% |
| 3rd | 22.4% | 21.3% | 23.9% |
| 4th | 3.8% | 3.1% | 4.7% |
| **Total sample** | 558 | 324 | 234 |
| | | (58%) | (42%) |

### 3.4. Method of Analysis

To analyze the proposed theoretical model and test the hypotheses, the Partial Least Squares (PLS-SEM) technique was used, with the Smart PLS software v.3.3.2 [66]. The analysis of the measurement model involved the reliability and validity of the constructs, as well as the structural model through $R^2$, the path coefficients, the confidence intervals, and the values of the Standardized Root Mean Square (SRMR) as a measure of approximate fit of the model for PLS-SEM [67]. A Common Method Bias (CMB) assessment was also performed.

Likewise, to identify the differences between social entrepreneurial intention before and during the COVID-19 crisis, a Student's *t*-test was carried out for differences in the means of construct values.

## 4. Results

### 4.1. Descriptive Analysis

From a descriptive analysis of the results, it can be seen (Table 3) that there is a clear social entrepreneurial intention on the part of the investigated population, with the mean of the items of the construct being slightly above the midpoint of the scale, between 4.50 and 4.78 (scale from 1 to 7).

The subjective norms or level of perceived support of the social environment for social entrepreneurial intention is high, with the items of this latent variable being between 5.54 and 5.93. Table 3 shows the results of the descriptive analysis (mean and standard deviation) of the items of the constructs of the proposed model. Social entrepreneurship attitude is also above the midpoint of the scale, and the average of the items is between 4.66 and 5.17. Similarly, perceptions of self-capacity and competencies (perceived behavioral control) to implement social entrepreneurship initiatives are

high, with the indicators measured for this latent variable being between 4.72 and 5.23, always above the midpoint of the scale, which ranges from 1 to 7.

**Table 3.** Descriptive analysis.

| | Constructs and Associated Items | Mean | Standard Deviation |
|---|---|---|---|
| **SEI** | **Social Entrepreneurial Intention** | | |
| | Indicate your level of agreement with the following sentences: | | |
| SEI1 | I am willing to do anything to start a social project. | 4.68 | 1.336 |
| SEI2 | My professional goal is to become a promoter of social projects. | 4.50 | 1.424 |
| SEI3 | I am determined to create a social project in the future. | 4.78 | 1.335 |
| **SEA** | **Social Entrepreneurship Attitude** | | |
| | Indicate your level of agreement with the following sentences: | | |
| SEA1 | Being a social entrepreneur has more advantages than disadvantages for me. | 4.66 | 1.283 |
| SEA2 | A career as a promoter of social projects is attractive to me. | 4.76 | 1.366 |
| SEA3 | Promoting social projects would be a great satisfaction for me. | 5.17 | 1.285 |
| **SN** | **Subjective Norms** | | |
| | If you decided to create a social project, would people in your close environment approve of that decision? | | |
| SN1 | Your closest family. | 5.93 | 1.270 |
| SN2 | Your friends. | 5.93 | 1.109 |
| SN3 | Your study partners. | 5.54 | 1.259 |
| **PBC** | **Perceived Behavioral Control** | | |
| | To what extent do you agree with the following statements regarding your entrepreneurial abilities? | | |
| PBC1 | Identify new opportunities. | 5.05 | 1.121 |
| PBC2 | Create new products and services. | 4.73 | 1.169 |
| PBC3 | Apply my personal creativity. | 5.23 | 1.194 |
| PBC4 | Be a leader and communicator. | 5.05 | 1.305 |
| PBC5 | Create a network of professional contacts. | 4.72 | 1.148 |
| PBC6 | Successfully administer/manage a project. | 5.11 | 1.110 |

Scale 1 to 7 (1 = Strongly disagree to 7 = Strongly agree).

### 4.2. Assessment of the Global Model

The results revealed SRMR model fit values of 0.058, with values lower than 0.08 being considered acceptable for PLS-SEM [67].

Additionally, the CMB has been used along with Harman's single-factor approach [68]. A CMB is present if a single or general factor seems to represent the majority of the variance. A non-rotational factor analysis using the eigenvalue criterion greater than one revealed three different factors that represented 63.9 percent of the variance. The first factor captured 36.9 percent of the variance in the data. Since no single factor emerged and the first factor did not account for most of the variance, the CMB does not appear to be a problem.

It has also been verified that there are no indications of multicollinearity between the antecedent variables of each of the endogenous constructs since all the VIF (Variance Inflation Factor) values are less than 5.

### 4.3. Measurement Model Assessment

The individual reliability of the indicators of the constructs, formulated in the reflective Mode A, is assessed by examining the loadings (λ) of the indicators with their respective construct. As shown in Table 4, all item loadings in the final measurement model are greater than 0.707 [69]. In Table 4, the reliability of the construct is analyzed, and it is observed how all the values of Cronbach's Alpha and of the composite reliability [70] are above the minimum cut-off point of 0.70 [71].

**Table 4.** Assessment results of the measurement model.

| | Constructs and Associated Items | Loading | Cronbach's Alpha | Composite Reliability | Average Variance Extracted (AVE) |
|---|---|---|---|---|---|
| | **Social Entrepreneurial Intention** | | **0.893** | **0.934** | **0.824** |
| SEI1 | I am willing to do anything to start a social project. | 0.907 | | | |
| SEI2 | My professional goal is to become a promoter of social projects. | 0.908 | | | |
| SEI3 | I am determined to create a social project in the future. | 0.908 | | | |
| | **Social Entrepreneurship Attitude** | | **0.853** | **0.911** | **0.773** |
| SEA1 | Being a social entrepreneur has more advantages than disadvantages for me. | 0.848 | | | |
| SEA2 | A career as a promoter of social projects is attractive to me. | 0.910 | | | |
| SEA3 | Promoting social projects would be a great satisfaction for me. | 0.878 | | | |
| | **Subjective Norms** | | **0.823** | **0.895** | **0.739** |
| SN1 | Your closest family. | 0.845 | | | |
| SN2 | Your friends. | 0.910 | | | |
| SN3 | Your study partners. | 0.821 | | | |
| | **Perceived Behavioral Control** | | **0.836** | **0.878** | **0.547** |
| PBC1 | Identify new opportunities | 0.815 | | | |
| PBC2 | Create new products and services. | 0.799 | | | |
| PBC3 | Apply my personal creativity. | 0.719 | | | |
| PBC4 | Be a leader and communicator. | 0.701 | | | |
| PBC5 | Create a network of professional contacts. | 0.734 | | | |
| PBC6 | Successfully administer/manage a project. | 0.714 | | | |

All latent variables achieve convergent validity since their AVE measurements exceed the level of 0.5 [71]. The discriminant validity was assessed by using the recommended approach of Fornell and Larcker [71] and examining the heterotrait-monotrait (HTMT) of the correlations, which is considered a stricter criterion [72].

The results in Table 5 show that the constructs examined exceeded the requirements of Fornell and Larcker [71] since all the correlations were less than the square of the AVEs and also the heterotrait-monotrait (HTMT) of the correlations (values less than 0.85 [73], which is considered a stricter criterion [72]). Therefore, the measurement model was considered satisfactory and provided sufficient evidence in terms of reliability and convergent and discriminant validity.

**Table 5.** Result of discriminant validity.

| Constructs | SEA | PBC | SN | SEI |
|---|---|---|---|---|
| **Fornell–Larcher [69]** | | | | |
| SEA | **0.879** | | | |
| PBC | 0.287 | **0.739** | | |
| SN | 0.365 | 0.153 | **0.860** | |
| SEI | 0.727 | 0.349 | 0.358 | **0.908** |
| **Heterotrait-Monotrait Ratio (HTMT)** | | | | |
| SEA | | | | |
| PBC | 0.334 | | | |
| SN | 0.434 | 0.174 | | |
| SEI | 0.829 | 0.389 | 0.418 | |

Note: The square root of AVEs is shown diagonally in bold.



### 4.4. Structural Model Assessment

The path coefficients (standardized regression coefficients) show the estimates of the structural model relationships, that is, the hypothesized relationships between constructs. The significance of the effects was assessed by bootstrapping [74]. Since the hypotheses specify the direction of the relationship of the variables, a one-tailed Student's t-distribution with n-1 degrees of freedom, where n is the number of subsamples, was used. There were 5000 samples made [75] with the number of cases equal to the number of observations in the original sample. To assess the significance of the relationships, in addition to bootstrapping, confidence intervals were analyzed [76].

As can be seen in Table 6 and Figure 2, the personal attitude toward social entrepreneurship has the greatest significant relationship with social entrepreneurial intention (H1: β = 0.647, $p < 0.001$). Subjective norms (H2: β = 0.097, $p < 0.01$) and perceived behavioral control (H3: β = 0.147, $p < 0.001$) also have a positive and significant relationship with social entrepreneurial intention, although the latter relationship has lower direct influence.

**Table 6.** Results of hypothesis testing.

| | | Path Coefficient | Sig. | T Statistics | Confidence Intervals | Confidence Intervals Bias | Supported |
|---|---|---|---|---|---|---|---|
| Hypothesis 1 | SEA -> SEI | 0.647 | *** | 21.156 | [0.594; 0.695] | [0.594; 0.695] | Yes/Yes |
| Hypothesis 2 | SN -> SEI | 0.097 | ** | 2.96 | [0.042; 0.151] | [0.041; 0.150] | Yes/Yes |
| Hypothesis 3 | PBC -> SEI | 0.147 | *** | 4.271 | [0.092; 0.205] | [0.089; 0.202] | Yes/Yes |
| Hypothesis 4 | SN -> SEA | 0.356 | *** | 8.22 | [0.283; 0.428] | [0.279; 0.423] | Yes/Yes |
| Hypothesis 5 | SN -> PBC | 0.156 | *** | 3.357 | [0.084; 0.235] | [0.077; 0.228] | Yes/Yes |

Bootstrapping using 5000 subsamples one-tailed t Student: ns: non-significant; ** $p < 0.01$; *** $p < 0.001$; t (0.05; 4999) = 1.645; t (0.01; 4999) = 2.327; t (0.001; 4999) = 3.092; Confidence Intervals [5–95%].

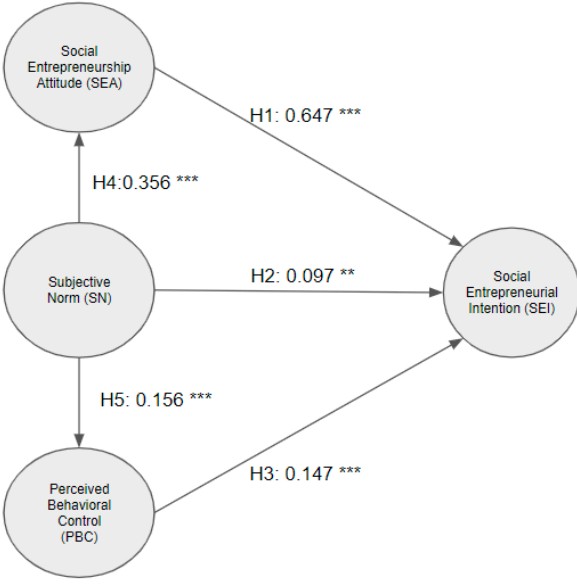

**Figure 2.** Results of analysis for social entrepreneurial intention. ns: non-significant; ** $p < 0.01$; *** $p < 0.001$.

The indirect relationship of subjective norms or social influence on social entrepreneurial intention was also tested through social entrepreneurship attitude (H4: β = 0.356, $p < 0.001$) and perceived behavioral control (H5: β = 0.156, $p < 0.001$) with indirect effects as seen in Table 7.

As stated above and in relation to the first aim of this work, to analyze the relationships between the variables identified as antecedents of social entrepreneurial intention, the hypotheses of the model are confirmed (H1, H2, H3, H4, and H5), although H2 is the weakest due to its low contribution and significance.

Therefore, we can affirm that there is a positive relationship between social entrepreneurship attitude, subjective norms, and perceived behavioral control and social entrepreneurial intention. In addition, there is a positive relationship between subjective norms and social entrepreneurial intention through the relationship with social entrepreneurship attitude and perceived behavioral control.

**Table 7.** Total, direct, and indirect effects.

|  | Direct Effects | Specific Indirect Effects | Total Effects |
|---|---|---|---|
| SEA -> SEI | 0.647 |  | 0.647 |
| PBC -> SEI | 0.147 |  | 0.147 |
| SN -> SEA -> SEI |  | 0.230 |  |
| SN -> PBC -> SEI |  | 0.023 |  |
| SN -> SEI | 0.097 |  | 0.350 |
| SN -> SEA | 0.356 |  | 0.356 |
| SN -> PBC | 0.156 |  | 0.156 |

The coefficient of determination ($R^2$) represents a predictive power measure that indicates the amount of variance of a construct that is explained by the predictor variables of the endogenous construct in the model. The proposed model explains 55.4% of social entrepreneurial intention, 12.7% of social entrepreneurship attitude, and 2.4% of perceived behavioral control (Table 8).

**Table 8.** Decomposition of variance, predictive relevance, and effect size.

|  | Path Coefficient | Variable Correlation | $R^2$ | $Q^2$ | $f^2$ |
|---|---|---|---|---|---|
| **Social entrepreneurship intention** |  |  | 55.4% | 0.451 |  |
| SEA -> SEI | 0.647 | 0.724 | 46.8% |  | 0.765 |
| SN -> SEI | 0.097 | 0.350 | 3.4% |  | 0.018 |
| PBC -> SEI | 0.147 | 0.352 | 5.2% |  | 0.044 |
| **Attitude toward social entrepreneurship** |  |  | 12.7% | 0.096 |  |
| SN -> SEA | 0.356 | 0.356 | 12.7% |  | 0.145 |
| **Perceived behavioral control** |  |  | 2.4% | 0.012 |  |
| SN -> PBC | 0.156 | 0.156 | 2.4% |  | 0.025 |

Effect $f^2$: <0.15 small; <0.35 moderate; ≥0.35 large.

Additionally, indicator $f^2$ (Table 8) evaluates the degree to which an exogenous construct contributes to explain a specific endogenous construct in terms of $R^2$ [77].

We observe that the relationship between subjective norms and entrepreneurial intention has a small effect size, thus its influence is limited. The relationships between subjective norms and perceived behavioral control, as well as between perceived behavioral control and social entrepreneurial intention, also have a small effect size.

On the other hand, as a criterion to measure the predictive relevance of the constructs, the Stone–Geisser test was used [78,79], observing in Table 8 that the $Q^2$ values are greater than zero, which indicates that the model has predictive potential.

### 4.5. Assessment of the Relationship between COVID-19 and Social Entrepreneurial Intention

To test the influence of the health crisis caused by COVID-19, a *t*-test was carried out for the mean differences of the items of the social entrepreneurial intention construct measured in two periods: before the crisis and during the crisis.

Table 9 shows that all the items that make up the latent variable have a significantly lower value during the crisis than before the crisis, thus it can be concluded that social entrepreneurial intention has decreased, confirming hypothesis 6.

**Table 9.** *t*-Test of the mean differences of the social entrepreneurial intention.

| Social Entrepreneurial Intention | Before (February) COVID-19 | During (June) COVID-19 | Dif. | Sig. | |
|---|---|---|---|---|---|
| I am willing to do anything to start a social project. | 4.78 | 4.55 | −0.24 | 0.039 | ** |
| My professional goal is to become a promoter of social projects. | 4.60 | 4.37 | −0.23 | 0.063 | * |
| I am determined to create a social project in the future. | 4.91 | 4.62 | −0.29 | 0.011 | ** |

Scale 1 to 7 (1 = Strongly disagree to 7 = Strongly agree). Level of significance: <0.05 **; <0.1 *; no significance "ns".

## 5. Discussion and Conclusions

This work analyzed, using the perspective of Ajzen's TPB [14], the relationship between the antecedent variables that make up this model and social entrepreneurial intention. Subsequently, the impact of the COVID-19 crisis on social entrepreneurial intention was measured.

From the analysis of the relationships between the variables considered as predictors of social entrepreneurial intention (social entrepreneurship attitude, subjective norms, and perceived behavioral control) and the performance of the behavior (social entrepreneurial intention), it is observed that hypotheses 1, 2, and 3 are fulfilled, with hypothesis 2, regarding the incidence of subjective norms, being the one with a weaker significance.

This assumes that there is a positive relationship between the social entrepreneurship attitude of university students [80] and their social entrepreneurial intention. In addition, there is a positive relationship between the perceived behavioral control of university students' ability to carry out social projects and their social entrepreneurial intentions. This confirms that the more positive the perceptions about one's own abilities are, the stronger the social entrepreneurial intention will be.

Regarding the weak relationship between university students' subjective norms and social entrepreneurial intention, this result coincides with that obtained in previous works by various authors such as Liñan and Chen [49], Autio et al. [46], and Krueger et al. [47], who used the TPB to measure the relationship between the variables that predict entrepreneurial intention. Liñan and Chen [49] suggest the non-significance of the relationship between subjective norms and entrepreneurial intention is due to the impact that this motivational factor has on this type of decision. In this case, it seems that the altruism that motivates the development of social and/or environmental projects [21] is more intense than the importance that the entrepreneurs themselves give to the perception of the opinions that their immediate environment has on the development of their project.

Hypotheses 4 and 5 were intended to measure the influence of subjective norms on social entrepreneurial intention through their relationship with social entrepreneurship attitude and perceived behavioral control. Both hypotheses are fulfilled, observing that there is more influence of subjective norms on social entrepreneurship attitude than on perceived behavioral control. This indicates that the perception of the opinions that people close to the individual have regarding the implementation of social and/or environmental projects affects more the attitude toward the behavior than the perception of their own ability and training for carrying out the entrepreneurial behavior.

The results that confirm hypotheses 1, 3, 4, and 5 of this work coincide with those obtained by Kruse [35]. This author analyzes, among other things, the direct and indirect effects on social entrepreneurial intention of antecedent variables according to Azjen's TPB [14]. In the case of hypothesis 2, which analyzes the impact of subjective norms on entrepreneurial intention, Kruse [35] obtains a non-significant result. Kruse's [35] study is applied to a total of 335 German promoters of social entrepreneurship projects. However, in the work of Tiwari et al. [48], in which Ajzen's TPB is applied to a sample of 390 students from the main technical universities in India, a positive impact, but of weak significance, is obtained for subjective norms in relation to entrepreneurial intention. These two different results show that the effect of the opinions from the environment (family, friends, colleagues) on social entrepreneurial intention is more relevant in people with entrepreneurial experience than in university students, with attitudes more prone to this type of initiative, in the line pointed out by Capella-Peris et al. [9].

Regarding the impact of the COVID-19 health crisis, it is observed that social entrepreneurial intention decreases after the pandemic. This result is explained to the extent that, following Kasych et al. [81], among the external barriers that exist linked to the development of social projects are those of an economic nature. It also coincides with the results obtained for traditional entrepreneurship in times of crisis (e.g., Devece et al. [55]) for which a clear negative impact is manifested.

In summary, and according to our results and also those obtained by Kruse [35], it seems that Ajzen's TPB [14] constitutes an ideal perspective to explain the formation of social entrepreneurial intention, taking into account the incidence of its antecedents, both directly and indirectly. Moreover, it is verified that the impact of subjective norms on entrepreneurial intention through social entrepreneurship attitude is more important than through perceived behavioral control.

In addition, despite the importance of promoting the creation of social entrepreneurship projects with the aim of providing innovative solutions to social and environmental problems, a situation of economic recession becomes a relevant barrier in the implementation of this type of enterprise.

In practice, the results obtained suggest the desirability of promoting the development of social entrepreneurship projects within the educational field, especially in university education, to the extent that at these ages, the promotion of social motivation may have the greatest impact. In this sense, it has been proven that the altruism associated with the social entrepreneurial intention of young people is more intense than the perceived opinions of their immediate environment on the development of their project.

The above coincides with what was pointed out by Tiwari et al. [48] in the sense that social entrepreneurship attitude is one of the variables that should be promoted in educational systems since its impact on entrepreneurial intention is greater than perceived behavioral control.

At the same time, it is logical to think that in an economic and social crisis climate, entrepreneurial intention decreases, since uncertainty generates a negative impact on the development of such intentions. In this sense, it would be interesting to develop educational actions that promote, especially among university students, the ability to identify entrepreneurial opportunities in the social field, even in times of economic crisis, taking into account that entrepreneurship by opportunity generates a greater impact in the long-term [56].

On the other hand, in the paper of Zaremohzzabieh [82], two alternative models were proposed and then evaluated, suggesting alternative formulations of the antecedents of social entrepreneurial intention by modifying the relationships between key TPB constructs and intentions. The findings revealed that the strength of the two models enriches the TPB through additional factors. This could be an upcoming challenge for a future extension of our work.

Fiore et al. [83] show the importance of creating teams with different competencies, cognitive and decision-making skills in entrepreneurship education. The creation of multidisciplinary teams could also be a good option for subsequent studies on social entrepreneurship.

Finally, it would be valuable to train university students in the ability to identify business opportunities despite possible situations of economic and social crises. This effort must be accompanied by public policies focused on facilitating the implementation of this type of initiative.

This study has certain limitations that open new research avenues. First, the sample used was made up of university students from one European country, as is commonly used in research into entrepreneurial intention, taking into account that higher education students could be included in the millennial generation, who share similar attitudes, perceptions, and experiences. Thus, having similar characteristics, it is possible to generalize the conclusions obtained [84]. However, to add more value to this line of research, it is proposed to extend the analysis carried out here to broader samples in order to test the model of formation of social entrepreneurial intention among students not only from other nationalities but from different academic fields and cultural backgrounds. The results obtained would also help personalize the training linked to the development of social entrepreneurship projects to obtain better results. Second, the research has been transversally designed, obtaining data from two periods: before the COVID-19 crisis and during the pandemic period. To develop

causal inferences, further empirical studies would be necessary that analyze the post-pandemic period. Finally, it would be useful to perform other studies including some control variables such as if students have previously participated in an entrepreneurship training course [85], entrepreneurial antecedents of their parents, etc.

**Author Contributions:** Conceptualization, I.R.-R. and F.J.G.-R.; methodology, I.R.-R., F.J.G.-R. and D.G.-T.; validation, D.G.-T.; formal analysis, I.R.-R., F.J.G.-R. and D.G.-T.; investigation, I.R.-R., F.J.G.-R. and D.G.-T.; resources, F.J.G.-R.; writing—original draft preparation, I.R.-R.; writing—review and editing, I.R-R. All authors have read and agreed to the published version of the manuscript.

**Funding:** This research received no external funding.

**Conflicts of Interest:** The authors declare no conflict of interest.

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
