# Peer review of "Social Entrepreneurial Intention and the Impact of COVID-19 Pandemic: A Structural Model"

_sustainability, doi:10.3390/su12176970_

Round 1

Reviewer 1 Report

Dear authors,

Fist of all I hope you all are fine in this complex period.

Thank you to give me the opportunity to read this study. I think it is analyzing an interesting topic.

The author(s) made a good effort to analyze the relationship between the Theory of Planned Behaviour and Social Entrepreneurial Intention. However, a major revision is required in order to improve the level of the paper and hopefully to improve readers to understand it and to cites it in the future. I put my comments in order and hope that the comments might be useful for you to enhance the manuscript. If you have any questions on these comments do not hesitate to contact me.

  1. Abstract

1.1 I suggest adding the year of the responses

1.2 I suggest reducing the length of the last sentence of the abstract. Maybe you can divide it into two sentences.

1.2 Maybe, but as you prefer, you can use introduce the abbreviation TPB for the Theory of Planned Behavior. You can do it also in the manuscript. As you prefer.

  1. Keywords:

I suggest improving the keywords by adding “Social entrepreneurship”

  1. Introduction

3.1 In the introduction I suggest adding a sentence about the Sustainable Development Goals (SDGs). Please take a look at this paper:

Horne, J., Recker, M., Michelfelder, I., Jay, J., & Kratzer, J. (2020). Exploring entrepreneurship related to the sustainable development goals-mapping new venture activities with semi-automated content analysis. Journal of Cleaner Production, 242, 118052.

3.2 Page 2 line 48:

I suggest adding that you apply this Theory to social entrepreneurial intention.

3.3 Page 2 from line 49 to line 54

I suggest adding the number of respondents from which university and the year of the survey

  1. Theory and hypothesis

4.1 Page 2 line 71

You write “The same author” but the reader cannot know who the author with the reference style in this journal is. So please write the name of the author. Thanks!

4.2 Page 2 line 77

I suggest always writing social and/or environmental.

4.3 Page 2 line 77

“This allows us to replicate in its,”. I suggest deleting it

4.4 Page 2 line 79

Before “For this reason, it is a priority...” I suggest adding a new sentence where you explain that since social entrepreneurs are facing social and/or environmental challenges, it is important to understand the variables relating to social entrepreneurial intentions in order to stimulate those variables.

Then you can write “For these reasons, it is a priority…”

4.5

I suggest adding something from this paper:

Zaremohzzabieh, Z., Ahrari, S., Krauss, S. E., Samah, A. A., Meng, L. K., & Ariffin, Z. (2019). Predicting social entrepreneurial intention: A meta-analytic path analysis based on the theory of planned behavior. Journal of Business Research, 96, 264-276.

4.6

I suggest adding related to the GUESSS project:  http://www.guesssurvey.org/

Please have a look at:

Fiore, E., Sansone, G., & Paolucci, E. (2019). Entrepreneurship education in a multidisciplinary environment: evidence from an entrepreneurship programme held in Turin. Administrative Sciences, 9(1), 28.

4.7 Page 2 from line 83 to 86

Sentence too long. Try to reduce it, thanks

4.8 Page 3 from line 95 to line 98

It is not clear here what social entrepreneur attitudes are. Can please you insert a more concrete example or please explain it better thanks!

4.9 Page 3 from line 101 to line 104.

Please can you something to explain in more detail what are social norms?

4.10 Page 3 line 205

Can you please add how and why the social norms impact entrepreneurial intentions from the literature you cited?

4.11

I do not agree that much that crises do not foster entrepreneurship. Can you please try to convince me a little more? Thanks!

  1. Empirical Study

5.1 Figure 1

5.1.1 Can you please add the “+” and the “-“ in the arrows based on your hypotheses?

5.1.2 Can you please add the abbreviations in all the circles? So it can be easier to read the tables with only abbreviations. Do not delete the long words, just adding the brackets with the abbreviations.

5.2 Page 5 line 188

I think you can say that the question was based on the literature and the GUESS project too but they are regarding social entrepreneurship.

5.3 Page 5

Please add the meaning from 1 to 7. In the answers, 1 was equal to? 7 was equal to? Thanks!

5.4 Table 1

5.4.1 SN please go more in detail in the questions regarding SN. The questions are not clear now. Thanks!

5.4.2 Please improve the text of SN in all the Tables

5.4.3 How did you define “social project” and “social entrepreneur” in the survey? Can you please add it to the manuscript? Maybe future studies can use your definition to perform similar studies and they will cite your paper.

5.4.4 I will suggest adding at the end of Table 1 something like The answers could range from 1 (meaning) to 7 (meaning)

5.5 Page 6 line 211

“March 21 and June 4, 2020.” Unfortunately, on June 4 we cannot say that it is “after COVID-19”. Therefore, my suggestion is to add a sentence here to explain here that it is a limitation of this study. Moreover, you can add it to the conclusion regarding the limit of this study.

5.6 Page 6 line 213

Can you please add more information regarding how did you perform the data collection? Who send the e-mails to all the students? Did you send just one e-mail or more than one?

5.7 Page 6 line 214

Can please add some information regarding the population? Then you need to say the response rate of 324 and the response rate of 234.

5.8 Table 2

5.8.1 Why you do not have answers from 5th year?

5.8.2 Can you please add some information about the studies of the students?

5.9 Page 7 line 228 and line 229

It is a general comment so please check it in all the manuscript. When you introduce an abbreviation, for instance: “standardized root mean square (SRMR)”:

  1. a) please introduce the abbreviation by using uppercase to the initial letter: Standardized Root Mean Square Residual (SRMR)
  2. b) after you introduce an abbreviation in the manuscript you have to use only the abbreviation.

This is true also to Common Method Bias (CMB), HTMT, etc..

  1. Results

6.1 Page 9 line 293

My suggestion is to not write “effect” or “impact” since you are doing a correlation analysis. I suggest always using “relationship” or “correlation”

6.2 Table 6

I do not understand why n = 5000 ?

6.3 Figure 2
As I suggest for Figure 1, can you please add the abbreviations in the circles?

6.4 Page 11 line 335 and 336

This is the most important comment: I am not sure that you can write that “so it can be concluded that social entrepreneurial intention has decreased, confirming hypothesis 6.” In more detail, if the sample who answered before the COVID-19 is different from the sample who answered “after” the COVID-19 then your results may be different mostly because the samples are different. Therefore, I do not know if you can test the hypothesis 6 whit your data. To do that you need to explain in detail that the first sample (before COVID-19) and the second sample (after COVID-19) are similar.

6.5 Table 9 line 338

Can you please add the meaning of 1 and 7 in the “Scale 1 to 7”?

  1. Discussion and conclusion

7.1 Page 12 line 371

Please try to explain in your opinion your results in practice

7.2 Page 12 from line 394 to line 396

I suggest not use the term “Second” and “Third” since in the manuscript you have always written you have two objects and not three. It can confuse the readers. My suggestion is to write something like Moreover, or Furthermore, or In addition to this,…

7.3

Is your study without limitations? :) I suggest adding some limitations and based on it suggest studies. E.g. this study is based only in one Country. Maybe other studies can analyze it in different Countries and maybe it will be possible to create a similar project to GUESSS but focused on Social Entrepreneurship. Moreover, you can suggest performing other studies by adding more control variables such as if they participate in an entrepreneurship course since Entrepreneurship Education is important to foster student entrepreneurship (e.g. Secundo et al., 2020), or if they have parents who created a startup etc etc..

Secundo, G., Mele, G., Sansone, G., & Paolucci, E. (2020). Entrepreneurship Education Centres in universities: evidence and insights from Italian “Contamination Lab” cases. International Journal of Entrepreneurial Behavior & Research.

Author Response

We would like to express our gratitude to the reviewer for the useful comments on our article. These suggestions have been considered carefully, which have allowed us to reflect more deeply on the issue of social entrepreneurial intention and the impact of COVID-19 pandemic, improving the content of the article. After making the suggested changes in the final version of the paper attached, we believe that it has been greatly improved in terms of its contribution to the academic field of this special issue. The final paper has not only been the result of the work of the authors but also due to the interesting contributions made by the reviewer to whom we reiterate our thanks. 

Reviewer 2 Report

Thank you for inviting me to do this review. This paper is novel in many respects. Firstly, the paper tackles an important topic in the literature which has not been addressed in previous studies. The study examines the impact of COVID 19 Pandemic on social entrepreneurial intention. Secondly, I believe the authors have an interesting data set collected in two phases. Finally, the empirical strategy is well articulated and findings well explained and linked to the contributions of the study.

Despite these good attributes, I would like to suggest some areas for improvement.

1.       The authors should mention/highlight the study context in the title or at least in the abstract.

2.       The introduction could benefit from a small statement on the organisation of the rest of the sections. Some readers are willing to know how the sections are organised in a paper.

3.       Consider using a proof-reader to read the paper as I saw a few minor typos and errors, for example, 90 entrepreneurial; 122 personal attitudes; H5: perceived

Thank you for the opportunity to review this paper. Overall, I am happy with the paper and hope it makes a significant contribution to the field of sustainability studies. I suggest very minor revisions as outlined in the comments above to make this paper publishable.

Author Response

(The authors gave the same response as above.)

Reviewer 3 Report

I think that the topic of the paper is interesting, but the authors have to increase their efforts in improving the contribution of the paper. In the current form, the paper appears to provide a limited contribution to the academic debate. For instance, from my point of view, it is already expected among researchers, practitioners and policy makers, that the COVID 19 negatively impacts on Social Entrepreneurial Intention of students. Thus the study should go beyond this prediction. Second, the  sample is limited to the students of a single university in a single European country. Finally, I suggest to the authors to improve significantly explanations about the implications of the study for the policy debate and business practice.
I hope that my suggestions will be useful to improve your study. Good luck for your research.

Author Response

(The authors gave the same response as above.)

Round 2

Reviewer 1 Report

Dear authors,

I hope you are fine in this complex period.

I want to say thank you for the edits you made to improve the manuscript. I think this will be the last major revision. After these changes, I will read again the all paper and I will inform you if there are any other minor changes you need to make. In any case, please read carefully the all revised manuscript before submitting it. My suggestion is to do contact an English expert to avoid any typos.

I am available for any clarifications if needed. 
Here you can find my suggestions:

  1. Introduction

1.1 Can you please define social entrepreneurship behavior after the sentence from line 24 to line 27?

1.2 Can you please divide the sentence from line 41 to line 43 into two sentences? If not, it is hard to understand. Thanks!

1.3 Line 56: can you please write “between crises and social entrepreneurial intentions” instead of “between crises and social entrepreneurship.”? Thanks!

1.4 Line 58: can you please write “practical implications” instead of “applied implications”? Please change it in all the manuscript

  1. Theory and hypotheses

2.1 Line 64: I suggest writing “TPB and Social Entrepreneurial Intention” instead of “Social Entrepreneurial Intention”

2.2 Line 65 and 66: in the sentence “There is a growing interest, both by academic and government institutions, in promoting social 65 and/or environmental entrepreneurship [16, 3, 11, 5].” I suggest also adding this work as a reference:

Sansone, G., Andreotti, P., Colombelli, A., & Landoni, P. (2020). Are social incubators different from other incubators? Evidence from Italy. Technological Forecasting and Social Change, 158, 120132. https://doi.org/10.1016/j.techfore.2020.120132

2.3 I do not like the sentence “Thus, while traditional entrepreneurship aims to obtain profits, social entrepreneurship relies on an altruistic attitude of the project promotion team [20, 11, 5].”. What do you mean with “project promotion team”? Can you please explain better it?

2.4 I do not like the sentence “This same author [24] introduces the term ‘sustainable social value’, precisely as a differentiating element between social entrepreneurship projects and actions related to charitable activities [24].”. Why you are associating social entrepreneur

2.5 Line 88: in the sentence “In short, social and/or environmental entrepreneurship projects are hybrid models [26, 1, 27, 18, 28] that function like traditional companies but incorporate an objective of a social and/or environmental nature.” I suggest also adding this work as a reference at the end of the sentence:

Leborgne‐Bonassié, M., Coletti, M., & Sansone, G. (2019). What do venture philanthropy organisations seek in social enterprises?. Business Strategy & Development, 2(4), 349-357. https://doi.org/10.1002/bsd2.66

2.6 H1: can you please justify a bit more why do you think that “There is a positive relationship between social entrepreneurship attitude and social entrepreneurial intention.”? In order to do this, I suggest adding a reference in the lines from 104 to 107 whit a reference to a study in order to explain that entrepreneurship attitude is positively related to entrepreneurial intention.

2.7 H3: it is not that clear. If someone will read this sentence of H3 it is hard to understand if perceived behavioural control impact the ability to carry out social projects and social entrepreneurial intention; or if perceived behavioural control and the ability to carry out social projects that impact social entrepreneurial intention. Can you please explain it better?

  1. Empirial Study

3.1 I suggest adding something related to the variables of COVID in the sentences: “That is why the research model includes four factors (see Fig. 1): social entrepreneurship attitude, subjective norm, perceived behavioral control and social entrepreneurial intention. Each factor was measured with multiple items. All items were adapted from extant literature to improve content validity.” I suggest this because in Fig. 1 there are five variables and not only four factors.

3.2 Line 210: I suggest writing “GEM report” instead of “this report”

3.3 Line 213: I suggest writing “for the development of social and/or environmental projects.” instead of “for the development of this type of projects.”

3.4 I do not understand why you put the sentence “Moreover, diverse studies have found out that young people in general and specifically higher education students could be included in Millennials generation, which are made up of people from 18 to 34 years old, who share similar attitudes, perceptions and experiences, having similar characteristics and thus being possible generalise the conclussions obtained [60].” into chapter 3.2. Measures…  I suggest moving in another part of the work or to delete it.

3.5 This is a very important comment: please ready very carefully the paper. There are some mistakes such as the words “Empirial Study” “conclussions.”. I really suggest an English revision of the paper.

3.6 I suggest writing everywhere “Strongly agree” instead of “Strongly Agree”

3.7 Line 225: I suggest deleting “of each subject” since you do not include all field of studies in the academia

3.8 I suggest moving the sentence “In this sense, Smith and Woodworth [63] recognize that a person with a high social entrepreneurial self-efficacy will tend to act more persistently in their goal of creating social value.” Into the Chapter “2. Theory and hypotheses”.

3.9 Table 1: Are you sure you asked this in the survey: “If you decided to create a firm, would people in your close environment approve of that decision?” you did not relate this question to “social project” instead of “firm”?

3.10 Table 1: Please delete the . into the sentence “To what extent do you agree with the following statements regarding your entrepreneurial capacity?.”

3.11 Line 253: I suggest defining better the field of studies. For instance, you can write Building and Civil Engineering. Moreover, I suggest defining better the field of studies related to Social Work and Industrial Relations. There are not clear.

3.12 Line 259: I suggest writing the dot before Therefore

3.13 Table 2, le percentages of the second column for the Degree studies do not sum to 100,00%. Please check all the numbers in the table. Thanks!

  1. Results

4.2 As I said in the first round of review, I suggest using the abbreviation for the common method bias in line 304.

4.3 Line 285: please delete the space after the (

4.4 Please make the same changes suggested before for the similar Table 1.

4.5 From line 346 to line 350: I suggest following the order of the hypotheses. Therefore, from H1 to H3.

4.6 Line 388: you are missing a dot at the end of the sentence

4.7 Line 415: you are missing a dot at the end of the sentence. Please read very carefully the paper.

  1. Discussion and conclusions

5.1 Line 514: please pay attention to how to cite papers in this journal. I think you need to cite in a different way the work of Gurtner and Soyez, (2016).

  1. References

6.1 Please read carefully all the references. There are some mistakes such as you are missing the number for the last citation.

Author Response

Response to Reviewer 1 Comments

We would like to express our gratitude to the reviewer for these second comments on our article. The suggestions have been considered carefully, which have allowed us to improve the content of the article. We respond to the reviewer comments in the order they were made in green.

Point 1. Introduction

1.1 Can you please define social entrepreneurship behavior after the sentence from line 24 to line 27?

Response: We have included a definition stating that social and environmental entrepreneurship is an enterprise project whose objective is to solve a social and/or environmental problem

1.2 Can you please divide the sentence from line 41 to line 43 into two sentences? If not, it is hard to understand. Thanks!

Response: It has been divided.

1.3 Line 56: can you please write “between crises and social entrepreneurial intentions” instead of “between crises and social entrepreneurship.”? Thanks!

Response: It has been corrected.

1.4 Line 58: can you please write “practical implications” instead of “applied implications”? Please change it in all the manuscript

Response: It has been corrected

Point 2: Theory and hypotheses

2.1 Line 64: I suggest writing “TPB and Social Entrepreneurial Intention” instead of “Social Entrepreneurial Intention”

Response: It has been changed

2.2 Line 65 and 66: in the sentence “There is a growing interest, both by academic and government institutions, in promoting social 65 and/or environmental entrepreneurship [16, 3, 11, 5].” I suggest also adding this work as a reference: Sansone, G., Andreotti, P., Colombelli, A., & Landoni, P. (2020). Are social incubators different from other incubators? Evidence from Italy. Technological Forecasting and Social Change, 158, 120132. https://doi.org/10.1016/j.techfore.2020.120132

Response: Thank you for the information regarding this interesting paper. We have read it and included the reference.

2.3 I do not like the sentence “Thus, while traditional entrepreneurship aims to obtain profits, social entrepreneurship relies on an altruistic attitude of the project promotion team [20, 11, 5].”. What do you mean with “project promotion team”? Can you please explain better it?

Response: It has been clarified.

2.4 I do not like the sentence “This same author [24] introduces the term ‘sustainable social value’, precisely as a differentiating element between social entrepreneurship projects and actions related to charitable activities [24].”. Why you are associating social entrepreneur

Response: It has been clarified.

2.5 Line 88: in the sentence “In short, social and/or environmental entrepreneurship projects are hybrid models [26, 1, 27, 18, 28] that function like traditional companies but incorporate an objective of a social and/or environmental nature.” I suggest also adding this work as a reference at the end of the sentence: Leborgne‐Bonassié, M., Coletti, M., & Sansone, G. (2019). What do venture philanthropy organisations seek in social enterprises?. Business Strategy & Development, 2(4), 349-357. https://doi.org/10.1002/bsd2.66

Response: Thank you for the reference to this relevant paper. We have read it and included the paper.

2.6 H1: can you please justify a bit more why do you think that “There is a positive relationship between social entrepreneurship attitude and social entrepreneurial intention.”? In order to do this, I suggest adding a reference in the lines from 104 to 107 whit a reference to a study in order to explain that entrepreneurship attitude is positively related to entrepreneurial intention.

Response: We have added some references about the positive relationship between attitude and entrepreneurial intention.

2.7 H3: it is not that clear. If someone will read this sentence of H3 it is hard to understand if perceived behavioural control impact the ability to carry out social projects and social entrepreneurial intention; or if perceived behavioural control and the ability to carry out social projects that impact social entrepreneurial intention. Can you please explain it better?

Response: The sentence of H3 has been changed.

H3: Perceived behavioral control positively influences social entrepreneurial intention.

Point 3: Empirial Study

3.1 I suggest adding something related to the variables of COVID in the sentences: “That is why the research model includes four factors (see Fig. 1): social entrepreneurship attitude, subjective norm, perceived behavioral control and social entrepreneurial intention. Each factor was measured with multiple items. All items were adapted from extant literature to improve content validity.” I suggest this because in Fig. 1 there are five variables and not only four factors.

Response: It has been corrected.  The research model includes five factors (see Fig. 1): social entrepreneurship attitude, subjective norms, perceived behavioral control, social entrepreneurial intention and crisis variable COVID-19

3.2 Line 210: I suggest writing “GEM report” instead of “this report”

Response: It has been corrected.

3.3 Line 213: I suggest writing “for the development of social and/or environmental projects.” instead of “for the development of this type of projects.”

Response: It has been corrected.

3.4 I do not understand why you put the sentence “Moreover, diverse studies have found out that young people in general and specifically higher education students could be included in Millennials generation, which are made up of people from 18 to 34 years old, who share similar attitudes, perceptions and experiences, having similar characteristics and thus being possible generalise the conclussions obtained [60].” into chapter 3.2. Measures…  I suggest moving in another part of the work or to delete it.

Response: It has been deleted.

3.5 This is a very important comment: please ready very carefully the paper. There are some mistakes such as the words “Empirial Study” “conclussions.”. I really suggest an English revision of the paper.

Response: The paper has been revised for an English language expert

3.6 I suggest writing everywhere “Strongly agree” instead of “Strongly Agree”

Response: It has been corrected.

3.7 Line 225: I suggest deleting “of each subject” since you do not include all field of studies in the academia

Response: It has been corrected.

3.8 I suggest moving the sentence “In this sense, Smith and Woodworth [63] recognize that a person with a high social entrepreneurial self-efficacy will tend to act more persistently in their goal of creating social value.” Into the Chapter “2. Theory and hypotheses”.

Response: It has been moved.

3.9 Table 1: Are you sure you asked this in the survey: “If you decided to create a firm, would people in your close environment approve of that decision?” you did not relate this question to “social project” instead of “firm”?

Response: It has been corrected.

3.10 Table 1: Please delete the . into the sentence “To what extent do you agree with the following statements regarding your entrepreneurial capacity?.”

Response: It has been deleted

3.11 Line 253: I suggest defining better the field of studies. For instance, you can write Building and Civil Engineering. Moreover, I suggest defining better the field of studies related to Social Work and Industrial Relations. There are not clear.

Response: It has been clarified.

3.12 Line 259: I suggest writing the dot before Therefore

Response: It has been corrected

3.13 Table 2, le percentages of the second column for the Degree studies do not sum to 100,00%. Please check all the numbers in the table. Thanks!

Response: The numbers have been corrected

Point 4: Results

4.2 As I said in the first round of review, I suggest using the abbreviation for the common method bias in line 304.

Response: It has now been used.

4.3 Line 285: please delete the space after the (

Response: It has been deleted

4.4 Please make the same changes suggested before for the similar Table 1.

4.5 From line 346 to line 350: I suggest following the order of the hypotheses. Therefore, from H1 to H3.

Response: The order has been changed

4.6 Line 388: you are missing a dot at the end of the sentence

Response: This has been corrected

4.7 Line 415: you are missing a dot at the end of the sentence. Please read very carefully the paper.

Response: It has been corrected and the paper has been read carefully

Point 5: Discussion and conclusions

5.1 Line 514: please pay attention to how to cite papers in this journal. I think you need to cite in a different way the work of Gurtner and Soyez, (2016).

Response: It is a mistake and it has been corrected.

Point 6: References

6.1 Please read carefully all the references. There are some mistakes such as you are missing the number for the last citation.

Response: The references have been checked.

Reviewer 3 Report

I thank the authors for the work done. The paper is now improved since its initial submission.

Author Response

We would like to express our gratitude to the reviewer for the useful comments on
our article.